# RoboRAN: A Unified Robotics Framework for Reinforcement Learning-Based Autonomous Navigation

**Matteo El-Hariry**[*]                                    *matteo.elhariry@uni.lu*
*Space Robotics Research Group, SnT*
*University of Luxembourg*

**Antoine Richard**[*]                                     *antoine.richard@uni.lu*
*Space Robotics Research Group, SnT*
*University of Luxembourg*

**Ricard M. Castan**[*]                                    *ricard.marsal@uni.lu*
*Space Robotics Research Group, SnT*
*University of Luxembourg*

**Luis F. W. Batista**[*]                                  *luis.batista@gatech.edu*
*Georgia Tech-Europe*
*IRL2958 GT-CNRS, Metz, France*

**Cedric Pradalier**                                       *cedric.pradalier@georgiatech-metz.fr*
*Georgia Tech-Europe*
*IRL2958 GT-CNRS, Metz, France*

**Matthieu Geist**                                         *matthieu@earthspecies.org*
*Earth Species Project*

**Miguel Olivares-Mendez**                                 *miguel.olivaresmendez@uni.lu*
*Space Robotics Research Group, SnT*
*University of Luxembourg*

**Reviewed on OpenReview:** *https://openreview.net/forum?id=0wDbhLeMj9*

## Abstract

Autonomous robots must navigate and operate in diverse environments, from terrestrial and aquatic settings to aerial and space domains. While Reinforcement Learning (RL) has shown promise in training policies for specific autonomous robots, existing frameworks and benchmarks are often constrained to unique platforms, limiting generalization and fair comparisons across different mobility systems. In this paper, we present a multi-domain framework for training, evaluating and deploying RL-based navigation policies across diverse robotic platforms and operational environments. Our work presents four key contributions: (1) a scalable and modular framework, facilitating seamless robot-task interchangeability and reproducible training pipelines; (2) sim-to-real transfer demonstrated through real-world experiments with multiple robots, including a satellite robotic simulator, an unmanned surface vessel, and a wheeled ground vehicle; (3) the release of the first open-source API for deploying Isaac Lab-trained policies to real robots, enabling lightweight inference and rapid field validation; and (4) uniform tasks and metrics for cross-medium evaluation, through a unified evaluation testbed to assess performance of navigation tasks in diverse operational conditions (aquatic, terrestrial and space). By ensuring consistency between simulation and real-world deployment, RoboRAN lowers the barrier to developing adaptable RL-based

---

[*]Equal contribution

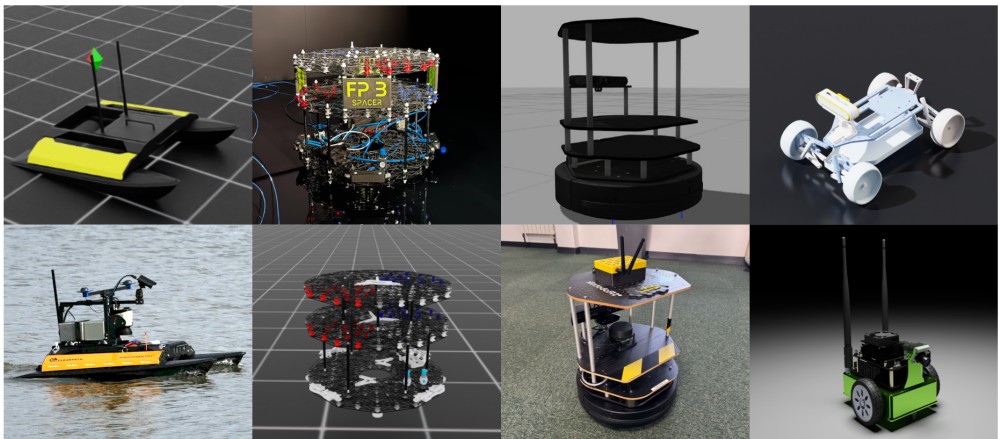

Figure 1: RoboRAN supports easy development of RL-based navigation tasks across a diverse set of robots. The five robots shown are Kingfisher, Floating Platform, Turtlebot2, Leatherback, and Jetbot. All have been implemented in simulation, while the first three have also been evaluated and demonstrated in real-world environments.

navigation strategies. Its modular design enables straightforward integration of new robots and tasks through predefined templates, fostering reproducibility and extension to diverse domains. To support the community, we release RoboRAN as open-source[1].

# 1 Introduction

One of the main goals of robotics research is to develop autonomous agents capable of navigating and executing tasks in any environment and medium, from terrestrial rovers and aquatic vessels to aerial drones and spacecraft. While Reinforcement Learning (RL) has shown promise in training such agents (Taheri et al., 2024; Wisniewski et al., 2024; Tezerjani et al., 2024; Surmann et al., 2020), more often than not, the capabilities of these agents are only illustrated on a single robot and on a single task. Existing benchmarks are often restricted to specific robot types and environment settings, limiting generalization and cross-domain comparisons. Although this approach is well suited to grow domain-specific knowledge, it makes it hard to see how one method would transfer to a different system. Hence, to scale RL for robotics, this paradigm must evolve toward more generalizable frameworks, enabling the development of universal solutions and their efficient adaptation to specific systems.

In the literature, we can find many specialized RL frameworks, be it used to control boats (Batista et al., 2024), satellites (El-Hariry et al., 2024; 2023), or ground rovers (Perille et al., 2020), but there has been little effort to evaluate these systems under a common structure. With this paper, we introduce RoboRAN, a multi-domain navigation framework focused on robot-task decoupling for simplifying the development and evaluation of RL-based policies across different robots and operational mediums. Our framework enhances Isaac Lab (Mittal et al., 2023), providing unified tasks and robots definition, and allowing to train one policy per robot-task pair using a shared and extensible training infrastructure. While supporting all the robots shown in Figure 1, RoboRAN offers the following four contributions:

1. **Modular design enabling task-robot interchangeability.** Our framework separates robot and task definitions using standardized APIs, minimizing integration overhead and enabling new robot–task pairings without altering existing modules. We demonstrate this interchangeability by training multiple navigation tasks across robots with distinct mobility systems (e.g., thrusters, wheels, water-based propulsion).

---

[1]https://github.com/snt-spacer/RoboRAN-Code

Table 1: Comparison of RoboRAN with existing RL frameworks.

| Benchmark | Domain Diversity | Task Types | Sim-to-Real | Robustness | Sensor / Env. Realism | Modularity | Backend |
|---|---|---|---|---|---|---|---|
| **RoboRAN (Ours)** | **Land / Water / Orbital** | **Navigation (4+)** | ✓ **(3 robots)** | **Partial** | **Moderate (realistic physics)** | ✓ | Isaac Lab |
| RL-Nav (Xu et al., 2023) (Xu et al., 2023) | Ground | Navigation (1) | (1 robot) | Partial | Moderate (realistic physics) | Partial | Gazebo |
| Habitat 2.0 (Szot et al., 2021) | Indoor | Rearrangement / Manipulation | — | Limited | High (photorealism, articulation) | Partial | Bullet |
| RRLS (Zouitine et al., 2024) | Sim (MuJoCo) | Continuous control | — | ✓ (worst-case) | Low | Moderate | MuJoCo |
| Robust Gymnasium (Gu et al., 2024) | Sim (varied tasks) | Control / Safe RL / Multi-agent | — | ✓ (disruptions) | Medium | ✓ | Gymnasium |
| FlightBench (Yu et al., 2024) | Aerial (quadrotors) | Ego-vision navigation | ✓ (1 robot) | Partial | High (occlusion, motion blur) | — | Custom |
| BARN (Perille et al., 2020) | Ground | Reactive / Safe navigation | ✓ (1 robot) | ✓ (safety/uncertainty) | Medium / Low | Partial | ROS / Gym |
| iGibson 0.5 (Xia et al., 2019) | Indoor | Interactive navigation | — | Limited | High (realistic sensors) | Partial | Gibson + PyBullet |
| Aquatic Benchmark (Corsi et al., 2024) | Water (aquatic) | Point-to-point navigation | — | Partial | Moderate (hydrodynamics, drift) | — | Unity3D |

2. **Sim-to-real transfer across diverse platforms with a single training and deployment pipeline.**
   In contrast to previous work that primarily validates policies on a single robot type, we conduct real-world experiments on three distinct platforms: a Floating Platform, a Boat, and a Turtlebot2. We demonstrate how our simulation platform, with domain randomization and physically grounded disturbances, enables sim-to-real transfer, and report the corresponding performance results across robot-task pairs.

3. **Open-source deployment API for Isaac Lab-trained policies.** We release the first deployment stack that enables policies trained with `skrl` or `rl_games` in Isaac Lab to run directly on real robots. The stack removes dependencies on the Gym wrapper, supports lightweight execution on edge devices, and provides ROS 2 integration and Dockerized workflows for reproducibility, lowering the barrier from training to real-world validation.

4. **Unified tasks and metrics for cross-medium evaluation.** We provide a common testbed for evaluating navigation strategies in a diverse set of operational conditions (e.g., terrestrial, aquatic, or microgravity environments). This is enabled by a reusable and extensible performance metric layer, applicable across navigation tasks and robot configurations.

## 2 Related Work

Reinforcement Learning (RL) has emerged as a powerful paradigm for control tasks, demonstrating its ability to learn complex policies directly from sensorimotor data. This has led to significant advancements across various domains, including robotic manipulation (Levine et al., 2015), humanoid locomotion (Peng et al., 2018), and the control of legged robots (Lee et al., 2020). While benchmark development has primarily centered on manipulation (Lee et al., 2019; James et al., 2019; Zhu et al., 2020; Heo et al., 2023), navigation remains a fundamental aspect of embodied intelligence that has gained increasing attention (Zhu & Zhang, 2021; El-Hariry et al., 2023).

To facilitate learning-based navigation, numerous simulation environments and physics engines have been developed. Frameworks such as MuJoCo (Todorov et al., 2012), PyBullet (Coumans & Bai, 2016), Webots (Michel, 2004), and Isaac Gym (Makoviychuk et al., 2021) provide efficient and scalable platforms for RL training, but are often constrained to single-domain settings or specific robot morphologies. Isaac Lab (Mittal et al., 2023) extends Isaac Gym by supporting diverse robotic platforms, though it lacks both a structured evaluation suite for benchmarking RL policies across tasks and domains and the flexibility for interchangeable training of multiple tasks across multiple robots. Several recent benchmarks have addressed learning-based navigation under specific environmental and sensory constraints. Habitat (Savva et al., 2019; Szot et al., 2021) targets high-level planning and mobile manipulation in photorealistic indoor environments. The BARN challenge (Perille et al., 2020) focuses on low-level control in cluttered scenes, while Flight-Bench (Yu et al., 2024) benchmarks ego-vision-based navigation for agile quadrotors. Aquatic navigation tasks are considered in (Corsi et al., 2024), and iGibson 0.5 (Xia et al., 2019) provides an interactive benchmark in household environments. These efforts, however, are typically domain-specific and lack support for robot–task interchangeability or sim-to-real evaluation.

Robustness and generalization have become important in recent benchmark development. RRLS (Zouitine et al., 2024) introduces worst-case robust control evaluation using adversarial domains in MuJoCo, while Robust Gymnasium (Gu et al., 2024) defines modular disruption models across 60+ tasks. Although these environments are well-suited to studying resilience in policy learning, they remain simulation-bound and are limited in the diversity of robotic embodiments.

Prior work such as (Xu et al., 2023) identifies four key desiderata for RL in robotics (uncertainty handling, safety guarantees, data efficiency, and generalization) and provides valuable evaluation metrics and insights. The Gazebo-based simulation environment used in this work supports algorithm comparisons, but is limited to a single navigation task and robot. In contrast, RoboRAN emphasizes multi-robot, multi-domain flexibility within a high-throughput, GPU-accelerated simulation stack. Its modular design and extensibility enable future integration of safety-focused features such as policy and environment constraints.

While robustness and safety-centric studies like RRLS (Zouitine et al., 2024), Robust Gymnasium (Gu et al., 2024), and (Xu et al., 2023) focus on domain shifts or guarantees, RoboRAN provides complementary value by supporting simple real-world deployment and modular task-robot definitions, allowing practitioners to easily integrate different robot morphologies and new navigation tasks across diverse physical environments. Table 1 compares RoboRAN with existing RL benchmarks along axes such as domain diversity, task types, sim-to-real validation, robustness testing, realism, and modularity. We highlight our benchmark's cross-domain reach, support for real-world deployment, and modular structure enabling extensibility.

## 3 RoboRAN Overview

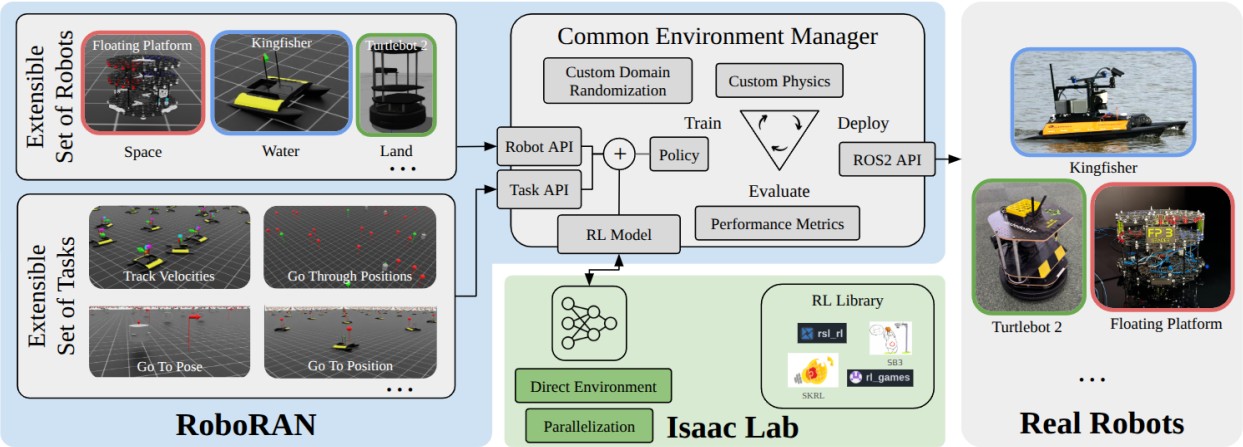

Figure 2: RoboRAN framework: the navigation tasks and Simulation Robots modules, along with a selected RL library of preference, serve as only inputs needed for our Environment Manager to train a policy in simulation, providing a readily available network for deployment on the real analog of the chosen robot.

RoboRAN is designed to train and evaluate robotic navigation tasks across a variety of operational settings. We introduce a unified structure where diverse robots can be evaluated on a shared set of tasks, using consistent interfaces and metrics. This design uniquely enables seamless interchangeability between agents and environments across different physical domains.

Our environment, formulated as a standard Markov Decision Process (MDP) (Puterman, 2014), is defined by the tuple $(\mathcal{S}, \mathcal{A}, P, r, \gamma)$, where $\mathcal{S}$ is the set of states, $\mathcal{A}$ is the set of actions, $P(s' \mid s, a)$ is the transition probability function, $r(s, a)$ is the scalar reward function, and $\gamma \in [0, 1]$ is the discount factor. At each time step $t$, an agent observes a state $s_t \in \mathcal{S}$, selects an action $a_t \in \mathcal{A}$ according to its policy $\pi(a_t \mid s_t)$, receives a reward $r_t = r(s_t, a_t)$, and transitions to a new state $s_{t+1} \sim P(\cdot \mid s_t, a_t)$. The environment thus provides at each step an observation $o_t \in \mathcal{O}$, a reward $r_t$, and a done signal $d_t \in \{0, 1\}$ indicating termination. The goal of the agent is to maximize the expected return $J(\pi) = \mathbb{E}_\pi \left[ \sum_{t=0}^{T} \gamma^t r(s_t, a_t) \right]$ over episodes of length $T$.

Figure 2 depicts the main components of our framework:

The **Common Environment Manager** instantiates a specified task-robot pair, dynamically configuring the simulation assets, physics parameters, and task constraints based on the robot's specific characteristics and operational medium. This modular design is a key contribution of our framework, as it enables full interchangeability between tasks and robots, and sub-module addresses a distinct aspect of this flexibility.

Table 2: Comparison of Robot Properties in RL Navigation Tasks. Control inputs are expressed as mathematical spaces.

| Robot | Actuation Type | Degrees of Freedom | Control Input Space | Motion Constraints |
|---|---|---|---|---|
| Floating Platform | Thruster-based (binary) | 3 (x, y, yaw) | $\{0, 1\}^8$ | No rolling/pitching, planar motion |
| Kingfisher | Water-based thrusters | 3 (x, y, yaw) | $\mathbb{R}^2$ (left/right thrust) | Drag and inertia effects, smooth but slow |
| Turtlebot2 | Differential drive | 3 (x, y, yaw) | $\mathbb{R}^2$ $(v, \omega)$ | No lateral movement, limited turn speed |

More details provided in Appendix A.7.

The **Custom Physics** module computes custom dynamics and actuation forces through parameterized thruster/propeller models. For instance, it applies hydrodynamic and propeller models for surface vessels, or microgravity and frictionless dynamics for the floating platform. It also enables flexible rewards to platform-specific constraints, such as penalizing rapid thruster actuation.

The **Custom Domain Randomization** module implements the disturbances detailed in Section 3.4 which are required to achieve sim-to-real transferability.

During training or evaluation, the *Performance Metrics* layer attach task-specific logging hooks enabling both on-line navigation metric updates and uniform post-hoc evaluation.

The **ROS2 API** simplifies policy deployment by using a ready-to-use inference node that exposes standard ROS2 interfaces, eliminating manual policy export steps that differ across RL libraries. More in the Deployment Section 3.6.

Together, these sub-modules enables flexibility and streamline the pipeline that shortens the loop between Training, Deployment and Simulation for diverse types of robots. The relationship between RoboRAN and Isaac Lab is detailed in Section 3.7.

## 3.1 Robots

RoboRAN supports all robots presented in Figure 1. Among them, we selected three representative robots for further evaluation and field tests. Their characteristics and control properties are summarized in Table 2.

**Land.** We selected the Turtlebot2, an open source platform with differential drive system with non-holonomic dynamics. To demonstrate the extensibility of our stack, two more wheeled robots (Leatherback and JetBot) are supported and tested in simulation. Results and details are provided in the Appendix A.1.

**Water.** We use the Kingfisher M200[2], a surface vessel with high inertial properties featuring a catamaran hull configuration and is driven by two fixed propellers, one on each hull. To simulate aquatic dynamics, we override Isaac Lab's default planar physics with custom hydrodynamics and hydrostatics models, enabling more accurate motion behaviors influenced by water resistance.

**Space.** We implement a floating platform, a thruster-actuated system constrained to planar movement, mimicking spacecraft-like motion with force-based control. This robot, through air bearings mounted on its base, generates a microgravity effect by pushing a constant airflow against the floor to lift and levitate in a free-floating fashion. To simulate this effect, we implement a custom frictionless dynamics to approximate free-floating orbital behavior, which is not natively supported by Isaac Lab.

## 3.2 Tasks

While Isaac Lab designs tasks around fixed robot models, RoboRAN decouples robot and task definitions, allowing consistent training and evaluation pipelines for any supported robot across all tasks. Our framework includes a suite of four navigation tasks designed to evaluate robotic motion in different environments and actuation methods. Each task leverages a structured observation space, detailed in Table 3, providing essential state information such as *base velocities* $[v_x, v_y, \omega]$, which capture the linear and angular velocity of the agent. To enhance temporal reasoning, we augment the observation vector with the previous action $a_{t-1}$, enabling the policy to infer dynamic transitions and improve stability in control. In all tasks, the observations are provided in the robot's own frame and apply Domain Randomization that mimics the noise of real sensors commonly used for state estimation.

---

[2]https://clearpathrobotics.com/

Table 3: Summary of Navigation Tasks, Objectives, and Observation Space

| Task | Objective | Obs Dim | Obs Components | Obs Variables |
|------|-----------|---------|----------------|---------------|
| GoToPosition | Reach a target position | 6 | Base Velocities, Target Info | $[v_x, v_y, \omega]$, $[d, \cos(\theta), \sin(\theta)]$ |
| GoToPose | Reach a target 3DoF pose | 8 | Base Velocities, Target Info, Target Heading | $[v_x, v_y, \omega]$, $[d, \cos(\theta), \sin(\theta)]$, $[\cos(\psi), \sin(\psi)]$ |
| GoThroughPositions | Follow a sequence of waypoints | $6 + 3n$ | Base Velocities, Target Info, Future Goals | $[v_x, v_y, \omega]$, $[d, \cos(\theta), \sin(\theta)]$, $[d_i, \cos(\theta_i), \sin(\theta_i)]$ |
| TrackVelocities | Maintain a set velocity | 6 | Error Terms, State | $[e_v, e_l, e_\omega]$, $[v_x, v_y, \omega]$ |

**GoToPosition** task requires the agent to reach a randomly initialized 2D position using the *target information* $[d, \cos(\theta), \sin(\theta)]$, representing the Euclidean distance and bearing to the goal. The relative angular position of the goal, is provided as a cos and sin of the angle to ensure the observations are continuous (Zhou et al., 2018).

**GoToPose** task is similar to *GoToPosition*, but also requires orientation alignment. Therefore, the observation space incorporates the *target heading* as $[\cos(\psi), \sin(\psi)]$ to provide the angular distance to the desired final orientation.

**GoThroughPositions** task involves sequential navigation through a series of $n$ waypoints, introducing *future goals* $[d_i, \cos(\theta_i), \sin(\theta_i)]$ in the observation space to ensure smooth trajectory planning.

**TrackVelocities** task requires the agent to follow a time-varying velocity reference in both linear and angular components. The observation space includes *velocity error terms* $[e_v, e_l, e_\omega]$ capturing deviations from the desired forward, lateral, and angular velocities. While no explicit path planner is embedded in the control policy, the velocity references can be derived from any arbitrary trajectory generator, including spline interpolators or MPC-based local planners. In this sense, the generator acts as a lightweight path planner, and the learned policy serves as a robust low-level controller that tracks planned motion commands across diverse robot morphologies and terrains.

These tasks provide a flexible evaluation suite for RL-based navigation, adaptable to use-cases such as autonomous docking, inspection, formation control, and trajectory tracking. While the core experiments in this paper focus on fundamental control-oriented tasks without obstacles or perceptual inputs, the framework is designed to support more complex scenarios. Thanks to its modular architecture, features such as obstacle avoidance, moving targets, and real-world sensing modalities can be integrated with minimal code changes. We provide illustrative examples of these extended capabilities, such as navigation with static obstacles, in Appendix A.2, demonstrating the framework's applicability beyond the tasks reported in the main evaluation.

### 3.3 Reward Formulation

The reward function combines task-specific objectives with general regularization terms to ensure consistent goal-directed behavior and control smoothness across robot types. Its unified form is shown in Eq. 1, where $d_p$, $d_h$, and $d_b$ denote the distance to the goal position, heading misalignment, and boundary proximity respectively. The terms $v_j$ represent linear and angular velocities clipped to task-defined ranges, $\Delta d_p$ is the signed progress along the goal direction, and $\mathbb{I}_{\text{goal}}$ provides a terminal bonus when the goal is reached. The term $r_t^{\text{robot}}$ adds optional robot-specific shaping such as control regularization.

$$r_t = \sum_{i \in \{p,h,b\}} w_i e^{-d_i/\lambda_i} + \sum_{j \in \{v,\omega\}} w_j \, \text{clip}(v_j, v_{\min}, v_{\max}) + w_{\text{pg}} \Delta d_p + w_{\text{bns}} \cdot \mathbb{I}_{\text{bonus}} + r_t^{\text{robot}} \tag{1}$$

The weights $w_i$, $w_j$, $w_{\text{pg}}$, and $w_{\text{succ}}$ vary by task, and are denoted in Table 10 as $\alpha_i$ for *GoToPosition*, $\beta_i$ for *GoToPose*, $\phi_i$ for *GoThroughPositions*, and $\gamma_i$ for *TrackVelocities* with decay constants $\lambda_i$ shared across tasks. For example, $\alpha_1 e^{-d_p/\lambda_1}$ encourages position convergence in *GoToPosition*, while $\beta_1 e^{-d_p/\lambda_1} e^{-d_h/\lambda_4}$ jointly rewards alignment in *GoToPose*. Similarly, progress is captured by $\phi_1 \Delta d_p$ in *GoThroughPositions*,

and *TrackVelocities* uses $\gamma_i e^{-e_i/\lambda_5}$ to penalize velocity tracking errors. All coefficients were tuned for balance and stability across robots, and are reported in Table 10 of the Appendix A.4 for full reproducibility.

### 3.4    Domain Randomization

To support sim-to-real transfer, we apply domain randomization in three key areas: (i) robot mass properties (mass, center of mass location, inertia tensor), (ii) actuation noise via Gaussian perturbations to commanded actions, and (iii) external disturbances modeled as random wrenches applied to the robot's base. The amount of randomization is chosen at random at every reset. We ensure reproducibility through a per-environment seed-controlled random number generation (RNG) using Warp (Macklin, 2022), allowing fine-grained domain randomization across parallel training environments. We apply moderate randomizations to simulate real-world uncertainties. For the Turtlebot2, we vary its mass by $\pm 0.1$ kg and CoM by $\pm 0.05$ m (std = 0.01), reflecting typical manufacturing variances. For the Kingfisher, which operates in a fluid environment, we use broader mass ($\pm 2.0$ kg) and CoM ($\pm 0.05$ m) perturbations, and apply random body wrenches (forces $\in [0, 0.25]$ N, torques $\in [0, 0.05]$ Nm) to account for water currents. For the Floating Platform, we use intermediate mass ($\pm 0.25$ kg) and similar CoM and wrench ranges to model small-scale system variations and external disturbances. An extensive description of the domain randomization is provided in the Appendix A.3.

### 3.5    Training

We train RL algorithms using the skrl (Serrano-Muñoz et al., 2023) library, with PPO (Schulman et al., 2017) as the training algorithm. PPO was selected due to its stability in high-dimensional continuous control and its widespread use in RL robotics settings. Rather than comparing algorithms, our focus is on demonstrating the decoupling of robot-task development within a unified framework. All experiments were run on a single NVIDIA RTX 4090. PPO was trained with default hyperparameters, listed in the Appendix A.5, and each robot-task pair converged in $\sim 15$ minutes on average. The final set of policies trained and used for evaluation are 12 (3 robots and 4 tasks).

### 3.6    Deployment

A key contribution of this work is the open-sourcing of the first deployment stack that seamlessly integrates policies trained with `skrl` or `rl_games` into the Isaac Lab simulation and real-robot environments. Our deployment framework is composed of three main components: (i) a *state creation node*, which constructs the observation vector required by the trained policy for a given robot-task configuration; (ii) a *model inference node*, which loads the exported policy checkpoint and publishes control commands; and (iii) a *goal generation node*, enabling dynamic task definition during execution.

To facilitate reproducibility and portability, the system is fully containerized with Docker, and offers direct ROS 2 integration for hardware deployment. Additionally, a simulated OptiTrack module allows rapid evaluation of inference pipelines without requiring physical experiments. Importantly, we deploy the system without relying on the Gym wrapper, which enables lightweight execution on edge devices without the need for workstation-grade resources to load the networks. This design ensures that controllers trained within Isaac Lab can be deployed to heterogeneous robotic platforms with minimal adaptation, making the stack suitable for both simulation benchmarking and field testing.

By open-sourcing this stack, we provide the community with the first unified and lightweight pathway from Isaac Lab training to real-world robotic control, lowering the barrier to reproducible RL deployment across platforms. Link to the code.[3]

### 3.7    Relation to Isaac Lab

While Isaac Lab provides a flexible starting point for robot simulation and training, RoboRAN extends it into a modular, benchmark-ready framework tailored for reinforcement learning. Our stack introduces (i) a

---

[3] https://github.com/snt-spacer/RoboRAN-deploy-to-robot

Table 4: **Task Success Criteria and Thresholds.** Each task defines success based on reaching position, orientation, velocity, or time-based constraints.

| Task | Success Condition | Threshold |
|---|---|---|
| GoToPosition | Final position error $\leq \epsilon_p$ | $\epsilon_p = 0.1m$ |
| GoToPose | $\epsilon_p$ and orientation error $\leq \epsilon_\theta$ | $\epsilon_p = 0.1m$, $\epsilon_\theta = 10°$ |
| GoThroughPositions | Waypoints reached within $\epsilon_{tp}$ | $\epsilon_{tp} = 0.2m$ |
| TrackVelocities | Maintain $\epsilon_v, \epsilon_w$ | $\epsilon_v = 0.2m/s$, $\epsilon_w = 10°/s$ |

decoupled robot–task interface enabling easy re-use of tasks across robots and vice versa, (ii) standardized reward APIs and evaluation metrics for consistent comparisons, (iii) a Dockerized ROS2 deployment interface for seamless sim-to-real transfer, and (iv) a suite of field-validated scenarios on heterogeneous platforms (ground, water, microgravity). These additions transform Isaac Lab from a development toolkit into a ready-to-use research benchmark for RL in mobile robotics.

# 4 Simulation Results

We evaluate our RL-trained policies in simulation across representative robot–task pairs, reporting results for three multi-domain robots: *Floating Platform*, *Kingfisher*, and *Turtlebot2*. These cover a diverse range of actuation models and navigation challenges, ensuring a broad evaluation scope.

## 4.1 Experimental Setup

Each policy is trained for 3200 epochs using PPO, over 5 random seeds per robot-task pair. During evaluation, we use GPU-accelerated Isaac Lab rollouts with parallel environments to collect performance data from 4096 evaluation episodes per run. All results are reported as mean $\pm$ std across environments. We define task-specific success as percentage of trajectories that satisfy the task specific metrics. Each metric is associated to a set of thresholds ($\epsilon_p$, $\epsilon_\theta$, $\epsilon_{tp}$, $\epsilon_v$, and $\epsilon_w$) that are listed in Table 4:
**GoToPosition**: distance to goal $< \epsilon_p$ within a fixed time budget.
**GoToPose**: both distance $< \epsilon_p$ and heading error $< \epsilon_\theta$ must be satisfied.
**GoThroughPositions**: count of waypoints reached in sequence within $\epsilon_{tp}$ tolerance before timeout.
**TrackVelocities**: mean absolute tracking error for linear and angular velocity must stay below $\epsilon_v$ and $\epsilon_w$.

In addition to success rate (defined as the percentage of episodes that meet task-specific thresholds), we report continuous evaluation metrics to capture control precision and stability:

**Final Distance Error (m)**: Euclidean distance to the goal at the end of the episode.
**Heading Error (°)**: Absolute orientation difference at the final timestep (GoToPose only).
**Time to Target (s)**: Duration required to reach the target precision threshold. Lower values reflect faster convergence.
**Velocity Tracking Error (m/s)**: Mean absolute error between target and actual linear/angular velocities (TrackVelocities only).
**Control Signal Variation (unitless)**: Standard deviation of control signals over the episode, reflecting smoothness or abruptness of control.
**Goals Reached**: Total number of intermediate targets successfully reached during sequential waypoint tasks (GoThroughPositions).

All these metrics are aggregated in Table 6, enabling a multi-dimensional comparison across tasks and robots.

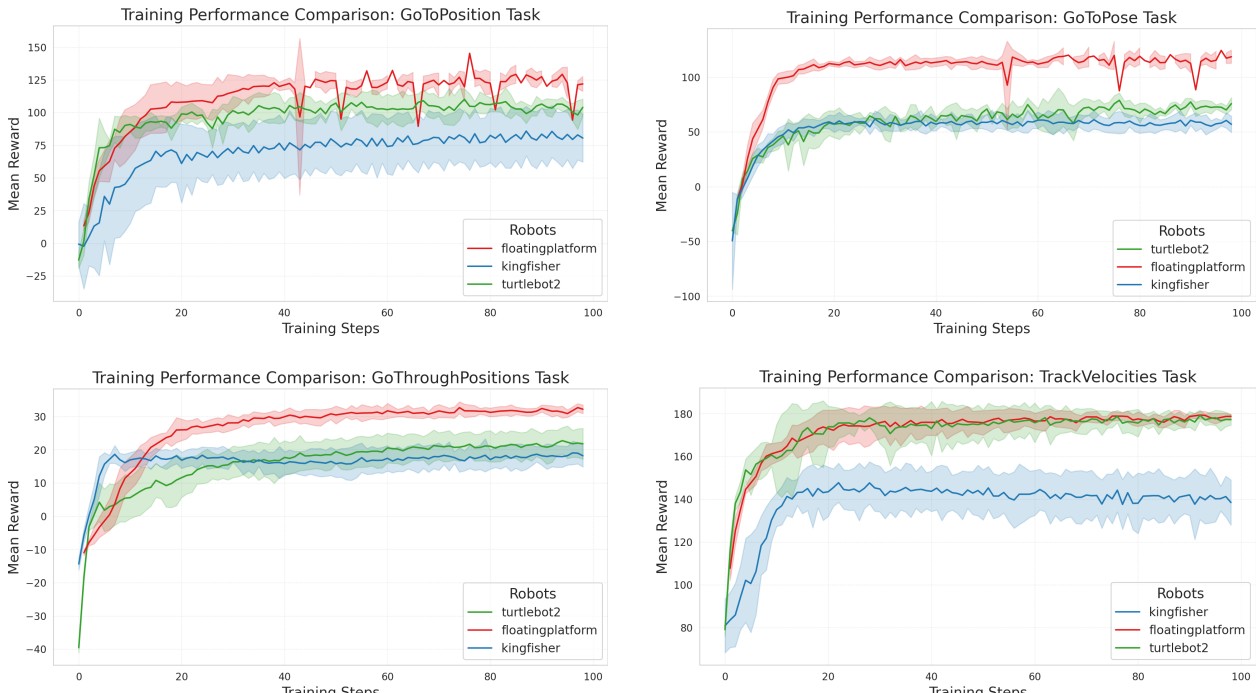

Figure 3: **Learning curves** showing rewards (mean ± std) over 5 seeds per robot, compared based on task.

Table 5: **Wall-clock time per robot-task pair (mean ± std over 5 seeds) in minutes [m].**

| Task | Floating Platform | Kingfisher | Turtlebot2 |
|------|-------------------|------------|------------|
| GoToPosition | $7.35 \pm 0.02$m | $13.91 \pm 0.19$m | $11.55 \pm 0.14$m |
| GoToPose | $5.46 \pm 0.00$m | — | $11.53 \pm 0.16$m |
| TrackVelocities | $5.08 \pm 0.18$m | $13.47 \pm 0.07$m | $11.28 \pm 0.02$m |
| GoThroughPositions | $5.51 \pm 0.13$m | $13.47 \pm 0.35$m | $11.20 \pm 0.03$m |

## 4.2 Training Efficiency and Learning Trends

Figure 3 shows the training reward across 5 seeds, highlighting learning speed and convergence per robot. The *FloatingPlatform* achieves the highest asymptotic rewards, benefiting from direct actuation despite its discrete thrust model. *Turtlebot2* converges reliably with moderate final returns, aided by low-dimensional control. *Kingfisher* shows slower and less stable learning, likely due to its hydrodynamic complexity and inertia. Table 5 reports average wall-clock time per training run. The *Kingfisher* requires the longest training time, consistent with its complex dynamics. *Turtlebot2* trains fastest among wheeled platforms. The unexpectedly short time for the *FloatingPlatform* suggests beneficial interaction between its discrete control structure and Isaac Lab's GPU-based parallelization. These differences motivate further study into simulation efficiency under varying robot dynamics.

## 4.3 Task Success and Performance Analysis

To complement the reward learning curves shown in Figure 3, we conduct a detailed quantitative evaluation across all robot-task pairs. This evaluation uses standardized success metrics and control efficiency indicators (Table 6) collected over 4096 parallel trajectories per setting.

Figure 4 presents the convergence curves for each robot-task pair. Shared tasks (*GoToPosition*, *GoThroughPositions*, and *TrackVelocities*) are plotted together for comparison, while specialized tasks (*GoToPose*) are shown separately.

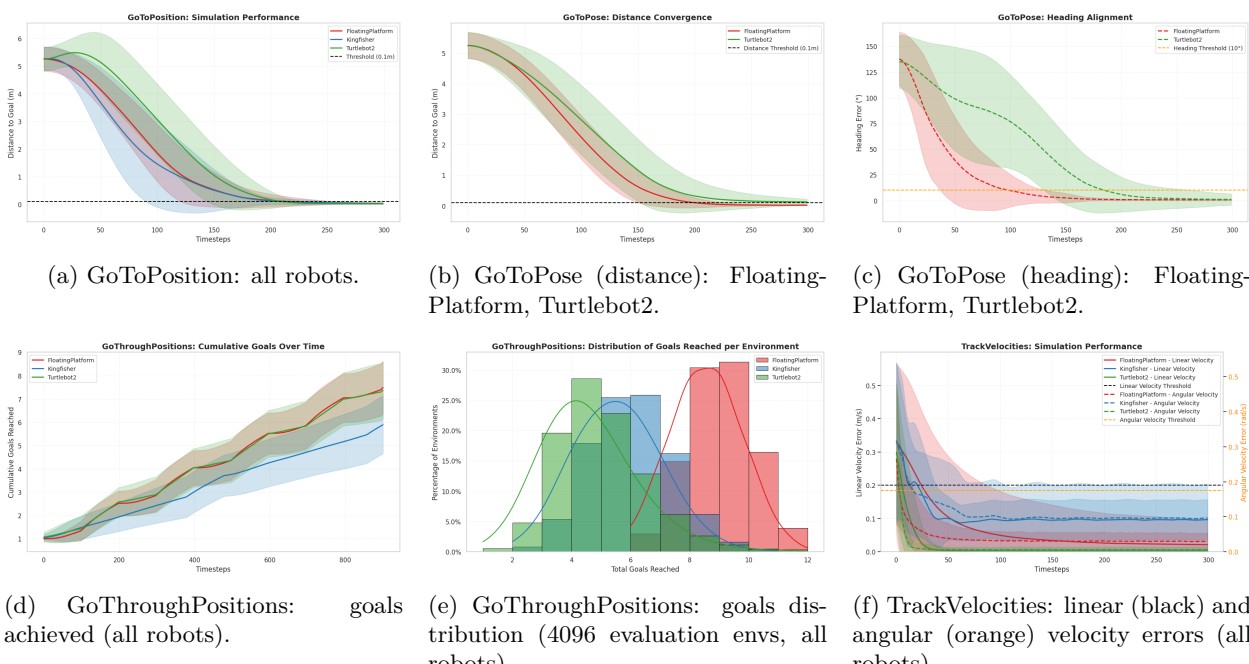

(a) GoToPosition: all robots.

(b) GoToPose (distance): Floating-Platform, Turtlebot2.

(c) GoToPose (heading): Floating-Platform, Turtlebot2.

(d) GoThroughPositions: goals achieved (all robots).

(e) GoThroughPositions: goals distribution (4096 evaluation envs, all robots).

(f) TrackVelocities: linear (black) and angular (orange) velocity errors (all robots).

Figure 4: **Simulation results across robots and tasks.** Performance comparisons for *GoToPosition*, *GoToPose*, *GoThroughPositions*, and *TrackVelocities* tasks. (a) All robots for GoToPosition. (b, c) Floating-Platform and Turtlebot2 on GoToPose (distance, heading). (d) Number of goals achieved in GoThroughPositions (all robots). (e) Goals distribution over 4096 parallel evaluation environments (all robots). (f) Linear velocity error in TrackVelocities (all robots).

Table 6: **Simulation evaluation metrics per task and robot (mean ± std across 4096 envs, PPO-skrl).** Metrics include: success rate (%), final distance error (m), heading error (°), time to target (s), velocity tracking error (m/s), control signal variation (unitless), and number of goals reached. "—" indicates non-applicable metrics.

| Task | Robot | Success Rate ↑ | Dist Err ↓ | Heading Err ↓ | Time to Target ↓ | Lin Vel Err ↓ | Ang Vel Err ↓ | Ctrl Var ↓ | Goals Reached ↑ |
|------|-------|----------------|------------|---------------|------------------|---------------|---------------|------------|-----------------|
| GoToPosition | FloatingPlatform | 0.94 ± 0.04 | 0.05 ± 0.01 | — | 87.05 ± 3.38 | — | — | 0.62 ± 0.04 | — |
| | Kingfisher | 0.59 ± 0.29 | 1.06 ± 0.73 | — | 176.11 ± 60.86 | — | — | 0.75 ± 0.45 | — |
| | Turtlebot2 | 0.99 ± 0.01 | 0.07 ± 0.00 | — | 92.60 ± 4.67 | — | — | 0.43 ± 0.26 | — |
| GoToPose | FloatingPlatform | 0.99 ± 0.01 | 0.02 ± 0.01 | 0.78 ± 0.01 | 92.38 ± 2.59 | — | — | 0.69 ± 0.05 | — |
| | Kingfisher | 0.66 ± 0.09 | 0.23 ± 0.06 | 7.07 ± 3.08 | 126.80 ± 31.29 | — | — | 0.48 ± 0.29 | — |
| | Turtlebot2 | 0.84 ± 0.04 | 0.14 ± 0.01 | 4.39 ± 1.56 | 131.49 ± 2.16 | — | — | 0.63 ± 0.38 | — |
| GoThroughPositions | FloatingPlatform | 1.00 ± 0.00 | 2.35 ± 0.25 | — | 65.18 ± 1.03 | — | — | 0.32 ± 0.04 | 13.57 ± 0.33 |
| | Kingfisher | 1.00 ± 0.00 | 2.41 ± 0.79 | — | 93.29 ± 18.56 | — | — | 0.43 ± 0.24 | 10.70 ± 2.84 |
| | Turtlebot2 | 1.00 ± 0.00 | 1.79 ± 0.05 | — | 101.50 ± 12.25 | — | — | 0.13 ± 0.06 | 11.01 ± 0.12 |
| TrackVelocities | FloatingPlatform | 0.93 ± 0.18 | — | — | — | 0.05 ± 0.07 | 0.03 ± 0.01 | 0.45 ± 0.04 | — |
| | Kingfisher | 0.48 ± 0.03 | — | — | — | 0.03 ± 0.00 | 0.24 ± 0.02 | 0.62 ± 0.37 | — |
| | Turtlebot2 | 0.77 ± 0.01 | — | — | — | 0.02 ± 0.01 | 0.11 ± 0.01 | 0.15 ± 0.09 | — |

**GoToPosition and GoToPose** The *Turtlebot2* achieves the highest success rate in *GoToPosition* with **0.99 ± 0.01**, benefiting from its differential-drive system and precise low-speed control. The *FloatingPlatform* follows with 0.94 ± 0.04, while the *Kingfisher* lags at 0.59 ± 0.29 due to inertia and limited turning agility. These trends are confirmed in Figure 4a, where Turtlebot2 reaches the goal region fastest, followed by FloatingPlatform and Kingfisher. In the *GoToPose* task, both *FloatingPlatform* and *Turtlebot2* succeed in reaching the target, with success rates of 0.99 ± 0.01 and 0.84 ± 0.04, respectively. *Kingfisher* is not evaluated due to its lack of heading control. FloatingPlatform achieves superior orientation control, with a heading error of **0.78° ± 0.01°**, compared to Turtlebot2's 4.39° ± 1.56°, as shown in Figure 4c. Distance convergence is also faster and more precise for FloatingPlatform (0.02 ± 0.01 m vs 0.14 ± 0.01 m, Fig. 4b).

**GoThroughPositions** All three robots successfully complete partial trajectories (100% success rate), but differ in the number of goals reached. FloatingPlatform achieves the highest average at **13.57 ± 0.33**,

while Turtlebot2 and Kingfisher reach **11.01 ± 0.12** and **10.70 ± 2.84** respectively. These differences are reflected in Figure 4d (cumulative goals) and Figure 4e (distribution), where *FloatingPlatform*'s performance is both higher and more consistent. Turtlebot2 shows smoother trajectories but fails to reach all waypoints within the time constraints, while Kingfisher's performance is more variable due to inertia limiting sharp turns.

**TrackVelocities**    *FloatingPlatform* demonstrates moderate success in tracking target velocities. Its linear velocity error is **0.05 ± 0.07**, and angular velocity error is **0.03 ± 0.01**, better than both Turtlebot2 (0.02 ± 0.01, 0.11 ± 0.01) and Kingfisher (0.03 ± 0.00, **0.24 ± 0.02**), as detailed in Table 6 and shown in Figure 4f. The high angular error for Kingfisher highlights the difficulty of fast heading corrections in water due to drag and momentum.

**Success Rate Summary**    Table 6 confirms these observations across tasks. Turtlebot2 dominates in *GoToPosition*, FloatingPlatform leads in *GoThroughPositions*, and both Turtlebot2 and FloatingPlatform perform comparably in *GoToPose*. In *TrackVelocities*, all robots achieve reasonable success, but Kingfisher exhibits the highest angular tracking errors, limiting its overall precision. These trends are visible in Figure 4, supporting our conclusion that control effectiveness varies not only across robots but also across tasks.

### 4.4   Discussions

While RL policies achieve high success rates, several robot-specific failure cases were observed. The *FloatingPlatform* experiences oscillations near target positions due to force-based control lag. The *Kingfisher* struggles with understeering in tight waypoint sequences, making sharp turns difficult. The *Turtlebot2*, despite overall fast learning, exhibits difficulty in precise in-place rotations, leading to longer turning maneuvers in the *GoToPose* task. These challenges highlight the need for refined reward shaping and constraint definitions to improve task execution. Overall, the successful training of diverse robots on shared tasks, despite their differing actuation and mobility constraints, demonstrates the viability of **unified cross-medium pipeline**. The *Turtlebot2*'s rapid convergence, the *FloatingPlatform*'s discrete thrust limitations, and the *Kingfisher*'s inertia-driven control difficulties highlight the importance of evaluating RL policies across heterogeneous platforms.

## 5   Sim-To-Real Results

### 5.1   Overview and Setup

We tested the trained policies on platforms (*FloatingPlatform*, *Kingfisher*, and *Turtlebot2*) across different navigation tasks. Real-world tests did not include the *GoToPose* task for the Kingfisher because wind, currents, and its non-holonomic motion often made goal poses unreachable. *TrackVelocities* was not reported for the Turtlebot as it directly executes velocity commands, making performance evaluation trivial. Each robot-task pair evaluated over 4 to 10 independent trials, with key metrics summarized in 7.

Sim-to-real performance drops are shown in Table 8. We omit results for the *GoThroughPositions* and *TrackVelocities* due to inconsistencies between simulation and real-world conditions. Specifically some task parameters (e.g. goal distances and target velocities) were adjusted to fit the physical testing areas, making direct comparison with simulation results unfair. A full breakdown of real-world evaluation runs, including per-trial plots and setup configurations, is available in Appendix A.6 and the public repository.

### 5.2   Field Results and Task Trends

**GoToPosition:** All three robots successfully minimized the distance-to-goal across runs, with Turtlebot2 and FloatingPlatform reaching perfect success rates of 1.00, and Kingfisher at 0.667. Despite Kingfisher's slower convergence and higher final distance error (0.464 ± 0.660), Turtlebot2 showed precise goal convergence (0.019 ± 0.044), matching simulation trends. Figure 5c illustrates these results.

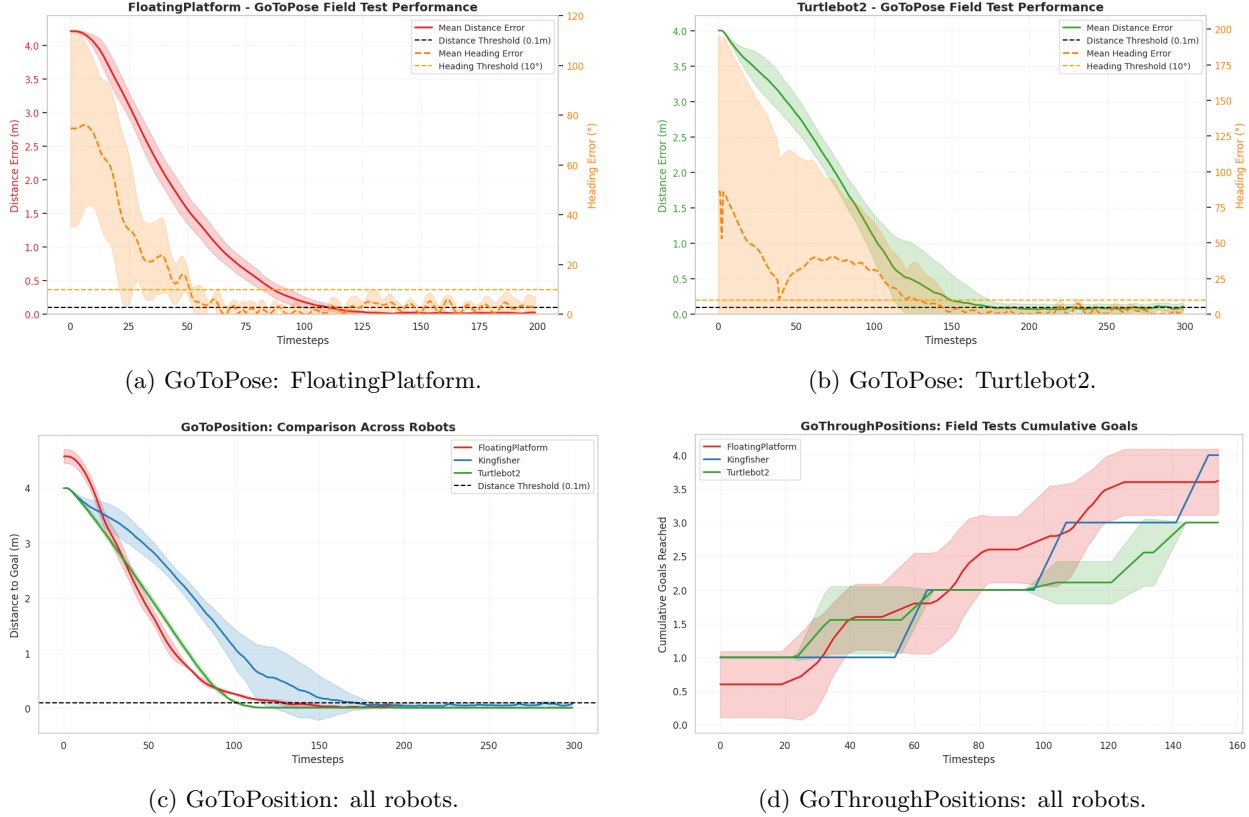

(a) GoToPose: FloatingPlatform.

(b) GoToPose: Turtlebot2.

(c) GoToPosition: all robots.

(d) GoThroughPositions: all robots.

Figure 5: **Field test results for navigation tasks.** Performance evaluation for *GoToPose* (FloatingPlatform, Turtlebot2), *GoToPosition* (all robots), and *GoThroughPositions* (all robots).

**GoToPose:** FloatingPlatform and Turtlebot2 both completed the task with high success rates (0.818 and 0.800, respectively), although heading alignment remained a challenge. FloatingPlatform achieved slightly better heading accuracy (3.68° ± 2.37°) than Turtlebot2 (4.21° ± 2.79°), as visualized in Fig. 5a–b.

**GoThroughPositions:** Although evaluated in the field, this task's structure changed significantly from simulation due to physical constraints. As such, we only report cumulative goals without comparison to sim results. FloatingPlatform consistently reached the most goals (6.00 ± 0.06), followed by Kingfisher (5.33 ± 1.03) and Turtlebot2 (4.60 ± 0.55), as seen in Fig. 5d.

**TrackVelocities:** We report how accurately each robot tracks time-varying linear and angular velocity commands. FloatingPlatform achieved the lowest angular tracking error (0.041 ± 0.019), benefiting from its symmetric thrust layout and rigid structure, while Kingfisher has a higher angular tracking error of 0.077 ± 0.049), consistent with the challenge of turning in water with drag-induced delay.

More results showing the repeatability of the field experiments are reported in Appendix A.6.

### 5.3 Sim-to-Real Comparison

To assess generalization, Table 8 compares simulation and real-world performance on shared metrics for GoToPosition and GoToPose. We observe modest drops in heading accuracy and final distance error, especially under noisy dynamics. However, success rates remain high, confirming strong policy transferability.

### 5.4 Discussions

Despite signals of good sim-to-real transfer, specific failure modes were observed. FloatingPlatform often overshot targets due to inertia and delayed braking. Kingfisher drifted in tight turns, likely due to hydrodynamic lag. Turtlebot2 struggled to rotate in place precisely, especially during heading alignment.

Table 7: **Real-world task performance** (mean ± std). "—" indicates not applicable or not tested.

| Task | Robot | Success | Final Dist Err | Heading Err | Ang Vel Err | Lin Vel Err | Goals Reached |
|------|-------|---------|----------------|-------------|-------------|-------------|---------------|
| GoToPosition | FloatingPlatform | 1.00 | 0.004 ± 0.009 | — | — | — | — |
| | Kingfisher | 0.667 | 0.464 ± 0.660 | — | — | — | — |
| | Turtlebot2 | 1.00 | 0.019 ± 0.044 | — | — | — | — |
| GoToPose | FloatingPlatform | 0.818 | 0.112 ± 0.061 | 3.68 ± 2.37 | — | — | — |
| | Turtlebot2 | 0.800 | 0.027 ± 0.020 | 4.21 ± 2.79 | — | — | — |
| GoThroughPos | FloatingPlatform | 1.00 | 0.112 ± 0.061 | — | — | — | 6.00 ± 0.06 |
| | Kingfisher | 1.00 | 0.235 ± 0.039 | — | — | — | 5.33 ± 1.03 |
| | Turtlebot2 | 0.00 | 0.287 ± 0.090 | — | — | — | 4.60 ± 0.55 |
| TrackVelocities | FloatingPlatform | — | — | — | 0.175 ± 0.027 | 0.056 ± 0.009 | — |
| | Kingfisher | — | — | — | 0.077 ± 0.049 | 0.041 ± 0.019 | — |
| | Turtlebot2 | — | — | — | — | — | — |

Table 8: **Sim-to-real performance gap** for shared metrics. Only GoToPosition and GoToPose are directly comparable.

| Task | Robot | Success (Sim → Real) | Δ Final Dist Err | Δ Heading Err |
|------|-------|----------------------|------------------|---------------|
| GoToPosition | FloatingPlatform | 0.94 → 1.000 | ↓ 0.05 → 0.014 | — |
| | Kingfisher | 0.59 → 0.667 | ↓ 1.06 → 0.460 | — |
| | Turtlebot2 | 0.99 → 1.000 | ↓ 0.07 → 0.019 | — |
| GoToPose | FloatingPlatform | 0.99 → 0.818 | ↑ 0.02 → 0.112 | ↑ 0.78° → 3.68° |
| | Turtlebot2 | 0.84 → 0.800 | ↑ 0.14 → 0.027 | ≃ 4.39° → 4.21° |

These observations suggest opportunities for improvement via adaptive feedback control, recurrent models, or further domain randomization, especially for tasks requiring fine heading convergence or dynamic stop conditions.

## 6 Conclusions

RoboRAN introduces a unified framework that decouples robot and task definitions, enabling reproducible training, evaluation and deployment of four navigation tasks on three heterogeneous robots spanning land, water, and microgravity. All robot–task pairs train in simulation within minutes and are assessed with consistent success metrics, facilitating direct comparison of learning progress and control efficiency across media. Policies learned in simulation transfer to the physical Turtlebot2, Kingfisher, and Floating Platform with high success rates and centimeter-level position errors, confirming effective sim-to-real generalization, though heading alignment and inertia-related failures highlight that the sim-to-real gap can be further minimized. Future work will extend RoboRAN to more complex tasks and additional robots, and its modular design provides a practical foundation for advancing robust autonomous navigation and studies of generalization across dynamics and platforms. In particular, we identify RoboRAN as a clean foundation for integrating transfer learning, multitask training, or policy distillation approaches.

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

# A   Appendix

## A.1   Wheeled robots

Along with the TurtleBot2, which we selected for real-world deployment and sim-to-real evaluation, RoboRAN also supports two additional wheeled robots in simulation: the **Leatherback**, NVIDIA's research platform for autonomous driving, and the **JetBot**, an open-source educational robot widely used for hands-on learning in robotics and AI.

The **TurtleBot2** and **JetBot** both rely on a differential drive system, enabling planar movement through independent control of left and right wheel velocities. This shared control structure supports consistent evaluation across these platforms. In contrast, the **Leatherback** features an Ackermann steering mechanism, better suited for high-speed and realistic road-like navigation. It offers advanced capabilities, including larger actuation limits and support for real-time onboard computation, making it a promising platform for future exploration of perceptually guided or high-speed navigation tasks. The **JetBot**, by comparison, is lightweight and cost-effective, making it ideal for rapid prototyping and student-level research.

While these two additional platforms were not included in our physical experiments, they are fully integrated into the RoboRAN framework and can be readily used to train and evaluate navigation tasks in simulation. Figure 6 shows their simulation models rendered within Isaac Lab.

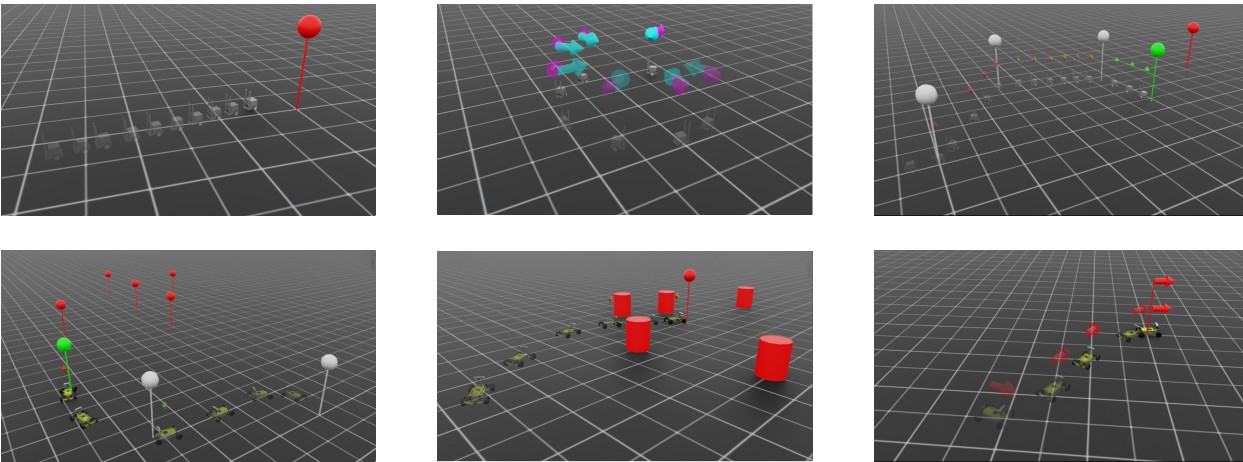

Figure 6: Left to right, first row Jetbot in GoToPosition, TrackVelocities, GoThroughPositions, second row Leatherback GoThroughPositions, GoToPositionWithObstacles, GoToPose

## A.2  Extended Environments and Task Variations

Although the four tasks described in the main paper are designed to evaluate core aspects of navigation, RoboRAN supports more complex environments and objectives. Its modular task interface allows users to easily introduce obstacles, moving targets, or perception-driven goals without altering the robot implementation or training pipeline.

To demonstrate flexibility under environmental constraints, we evaluate the *GoToPosition* task with static obstacles placed between the robot and its sequence of goals. Using the same reward structure and PPO hyperparameters as in the base task, without task-specific tuning, all three robots (FloatingPlatform, Kingfisher, Turtlebot2) successfully learn to navigate around obstacles (Fig. 8). Training curves (Fig. 7) show rapid improvement followed by stable convergence, with final mean rewards over the last 50 steps of $\sim 93.4$ (Kingfisher), $\sim 75.4$ (Turtlebot2), and $\sim 73.0$ (FloatingPlatform). The observation space is extended from the base task by appending the positions of the three closest objects in addition to the original six dimensions.

Additional complex scenarios, such as manipulation-inspired tasks (e.g., push-block for wheeled robots), are also supported but are left out of the main scope. These can be integrated with minimal code changes and will be shared in future iterations of the framework.

## A.3  Domain Randomization

We adopt a domain randomization framework as a standard tool to facilitate sim-to-real transfer, perturbing environment and robot parameters during training. The framework supports multiple application timings: reset-based (applied once at the start of an episode), step-based (applied at every simulation step), action-

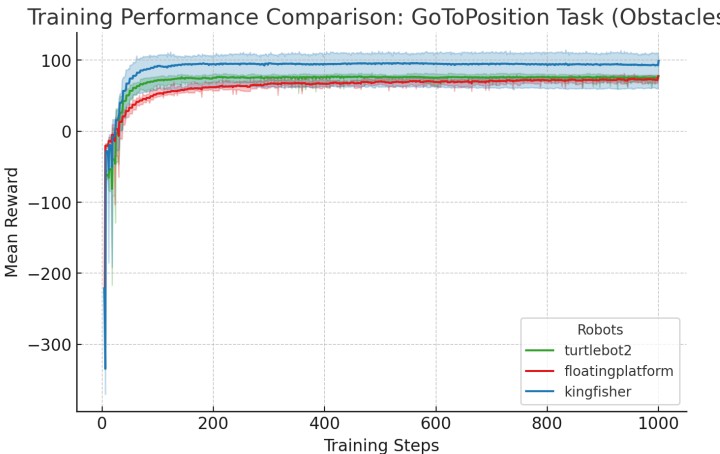

Figure 7: Training performance on *GoToPosition* with static obstacles for three robots. Curves show mean and std reward over 10 seeds. All robots learn stable obstacle-avoiding behaviors; the final-50-step mean rewards are **Kingfisher** $\approx 93.4$, **Turtlebot2** $\approx 75.4$, and **FloatingPlatform** $\approx 73.0$.

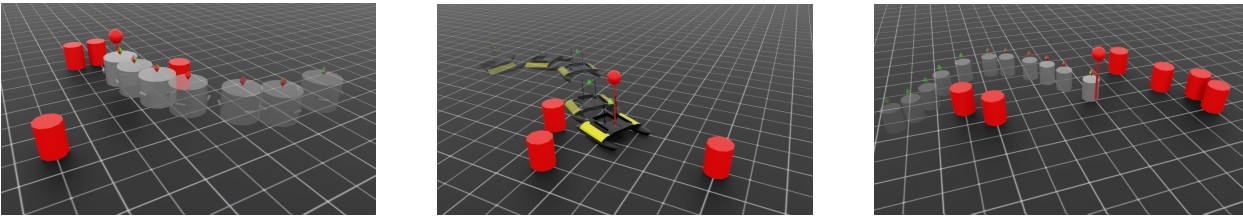

Figure 8: FloatingPlatform, Kingfisher, and Turtlebot2 in GoToPositionWithObstacles

based (applied when processing control actions), and observation-based (applied when processing sensory observations).

Table 9 summarizes the available randomization types, supported modes, application timings, and descriptions. These include variations of physical parameters such as mass, center of mass (CoM), and inertia, as well as noise and scaling in actions and observations, and exogenous disturbances in the form of wrenches. Both stochastic (uniform/normal) and deterministic (decay, sinusoidal) patterns are supported, and modes can be combined.

Each randomization category is parameterized to control its magnitude, distribution, affected elements, and temporal behavior. For example, *MassRandomizationCfg* can vary a body's mass according to a uniform or normal distribution, or apply gradual decay either over time or as a function of action magnitude. Similarly, *CoMRandomizationCfg* offsets the center of mass in two or three dimensions, and *InertiaRandomizationCfg* perturbs the inertia tensor.

The framework also supports stochastic perturbations at the action and observation interfaces. *NoisyActionsCfg* injects bounded uniform or Gaussian noise into control commands, while *ActionsRescalerCfg* applies multiplicative scaling to emulate actuator gain variability. *NoisyObservationsCfg* adds perturbations to sensed state variables, with optional normalization after noise injection.

External disturbances are modeled through *WrenchRandomizationCfg*, which can produce sporadic "kicks" or continuous wrenches, following uniform, Gaussian, or sinusoidal patterns. Parameters include affected body names, force and torque ranges, kick frequencies, and waveform characteristics.

The randomization system is implemented on top of a core class (RandomizationCore) that provides mode dispatch, timing control, environment integration, and logging facilities. Each specific randomization type inherits from this core, specializing the parameter perturbations and application rules. Finally, the

Table 9: Randomization types, modes, application timings, and descriptions.

| Type | Mode Name | Applied At | Description |
|------|-----------|------------|-------------|
| Mass | `uniform` | Reset | Uniform mass variation at episode start |
| Mass | `normal` | Reset | Normal-distributed mass variation at episode start |
| Mass | `constant_time_decay` | Step | Mass decays over time during an episode |
| Mass | `action_based_decay` | Step | Mass decays proportionally to action magnitude |
| CoM | `uniform` | Reset | Uniform CoM offset at episode start |
| CoM | `normal` | Reset | Normal-distributed CoM offset at episode start |
| CoM | `spring` | Step | Spring-like CoM behavior (placeholder) |
| Inertia | `uniform` | Reset | Uniform inertia variation at episode start |
| Inertia | `normal` | Reset | Normal-distributed inertia variation at episode start |
| Inertia | `decay` | Step | Inertia decays over time |
| Actions | `uniform` | Reset + Action | Uniform noise sampled on reset, applied during actions |
| Actions | `normal` | Reset + Action | Normal noise sampled on reset, applied during actions |
| Rescaler | `uniform` | Reset + Action | Uniform scaling factors sampled on reset, applied during actions |
| Observations | `uniform` | Reset + Observation | Uniform noise sampled on reset, applied during observations |
| Observations | `normal` | Reset + Observation | Normal noise sampled on reset, applied during observations |
| Wrench | `kick_uniform` | Reset + Step | Sporadic uniform-distributed force/torque disturbances |
| Wrench | `kick_normal` | Reset + Step | Sporadic normal-distributed force/torque disturbances |
| Wrench | `constant_uniform` | Reset + Step | Constant uniform-distributed wrenches applied every step |
| Wrench | `constant_normal` | Reset + Step | Constant normal-distributed wrenches applied every step |
| Wrench | `constant_sinusoidal` | Reset + Step | Sinusoidally varying wrenches applied every step |

framework remains compatible with perturbation utilities (e.g., rigid body material randomization, actuator gain variation, gravity changes) from Isaac Lab. Code 1 shows a sample configuration class combining multiple randomization types. In this example, mass is perturbed both at reset and via time decay; CoM is randomized uniformly at reset; action commands are both noised and rescaled; and sporadic force/torque kicks are applied to a designated body.

```
1  @configclass
2  class RobotCfg:
3      mass_rand_cfg = MassRandomizationCfg(
4          enable=True,
5          randomization_modes=["normal", "constant_time_decay"],
6          body_name="core",
7          max_delta=0.1,
8          mass_change_rate=-0.025
9      )
10     com_rand_cfg = CoMRandomizationCfg(
11         enable=True,
12         randomization_modes=["uniform"],
13         body_name="core",
14         max_delta=0.05
15     )
16     noisy_actions_cfg = NoisyActionsCfg(
17         enable=True,
18         randomization_modes=["uniform"],
19         slices=[(0, 2)],
20         max_delta=[0.025],
21         clip_actions=[(-1, 1)]
22     )
23     actions_rescaler_cfg = ActionsRescalerCfg(
24         enable=True,
25         randomization_modes=["uniform"],
26         slices=[(0, 2)],
27         rescaling_ranges=[(0.8, 1.0)],
28         clip_actions=[(-1, 1)]
29     )
30     wrench_rand_cfg = WrenchRandomizationCfg(
31         enable=True,
32         randomization_modes=["kick_uniform"],
33         body_name="core",
34         uniform_force=(0, 0.25),
35         uniform_torque=(0, 0.05),
36         push_interval=5
37     )
```

Code 1: Example multi-randomization configuration.

### A.4 Task specific reward parameters

Table 10: Reward parameters for PPO training. Task-specific coefficients and decay values used in Equation 1.

| GoToPosition | | GoToPose | | GoThroughPos. | | TrackVelocities | |
|---|---|---|---|---|---|---|---|
| $\alpha_{i1}$ (pos) | 1.0 | $\beta_{i1}$ (pose align) | 1.0 | $\phi_{i1}$ (progress) | 1.0 | $\gamma_{i1}$ (lin vel err) | −1.0 |
| $\alpha_{i2}$ (head) | 0.25 | $\beta_{j1}$ (lin vel) | −0.05 | $\phi_{i2}$ (head) | 0.05 | $\gamma_{i2}$ (ang vel err) | −0.5 |
| $\alpha_{j1}$ (lin vel) | −0.05 | $\beta_{j2}$ (ang vel) | −0.05 | $\phi_{j1}$ (lin vel) | 0.0 | $\gamma_{i3}$ (bonus) | 0.0 |
| $\alpha_{j2}$ (ang vel) | −0.1 | $\beta_{bns1}$ (boundary) | −10.0 | $\phi_{j2}$ (ang vel) | −0.05 | $\gamma_{bns1}$ (boundary) | −10.0 |
| $\alpha_{bns1}$ (bonus) | −10.0 | $\beta_{pg1}$ (progress) | 0.2 | $\phi_{bns1}$ (bonus) | −10.0 | — | — |
| $\lambda_1 = 1.0$ (dist) | | $\lambda_2 = 0.25$ (head) | | $\lambda_3 = 1.0$ (bnd) | | $\lambda_4 = 1.0$ (vel err) | |

## A.5   Algorithm hyperparameters

Table 11 summarizes the PPO hyperparameters used across all robot–task pairs. These values are derived from our default training configuration used in all experiments.

Table 11: PPO hyperparameters used in all experiments.

| Parameter | Value |
|---|---|
| Rollouts per update | 32 |
| Learning epochs per update | 8 |
| Mini-batches per epoch | 8 |
| Discount factor ($\gamma$) | 0.99 |
| GAE parameter ($\lambda$) | 0.95 |
| Learning rate | $5 \times 10^{-4}$ |
| Learning rate scheduler | KLAdaptiveLR (threshold $= 0.008$) |
| State/value preprocessors | RunningStandardScaler |
| Gradient norm clipping | 1.0 |
| Clipping ratio ($\epsilon$) | 0.2 |
| Value function clip | 0.2 |
| Clip predicted values | True |
| Value loss coefficient | 2.0 |
| Entropy loss coefficient | 0.0 |
| KL threshold (early stop) | 0.0 |
| Time-limit bootstrap | False |

## A.6   Field tests repeated experiments

In this section we provide additional details from the field trials. Figure 9 shows multiple trajectories for different combinations of robots and tasks, and Figure 10 provides additional metrics for the *GoToPosition* task across all three evaluated robots: FloatingPlatform, Kingfisher, and Turtlebot2. In these trials, observations are affected by onboard sensor noise and real-world environmental factors. The results highlight the consistency and cross-platform transferability of the learned policies under realistic operating conditions.

The **Turtlebot2** (Fig. 9a) exhibits consistent, nearly straight-line paths with minimal deviation across trials. The differential drive system enables precise low-speed control, resulting in rapid and stable goal convergence. The small variability between runs is attributed to minor noise in velocity execution and sensor estimates.

In the case of the **Kingfisher** (Fig. 9b), trajectories show larger variance and slower convergence, particularly in the final approach phase. This behavior reflects the platform's higher inertia and complex drag-dominated dynamics, which lead to increased difficulty in performing precise maneuvers, especially under subtle environmental disturbances such as wind or current. These results suggest opportunities to reduce the sim-to-real gap through improved dynamics modeling.

For the **FloatingPlatform** (Fig. 9c), trajectories remain tight and exhibit rapid convergence to the target position, with small lateral overshoots likely due to inertia and the discrete nature of thrust control. Despite minor oscillations near the goal, all runs demonstrate high repeatability and stable stopping behavior.

Across all platforms, the consistency of trajectory profiles is mirrored by relatively low action variability during the task, confirming that policies rely on smooth and robust control strategies even under unstructured real-world conditions. These results reinforce the generalization of our learned policies in real-world deployments across different mobility systems, especially when operating under noisy sensing, actuation, and environmental disturbances.

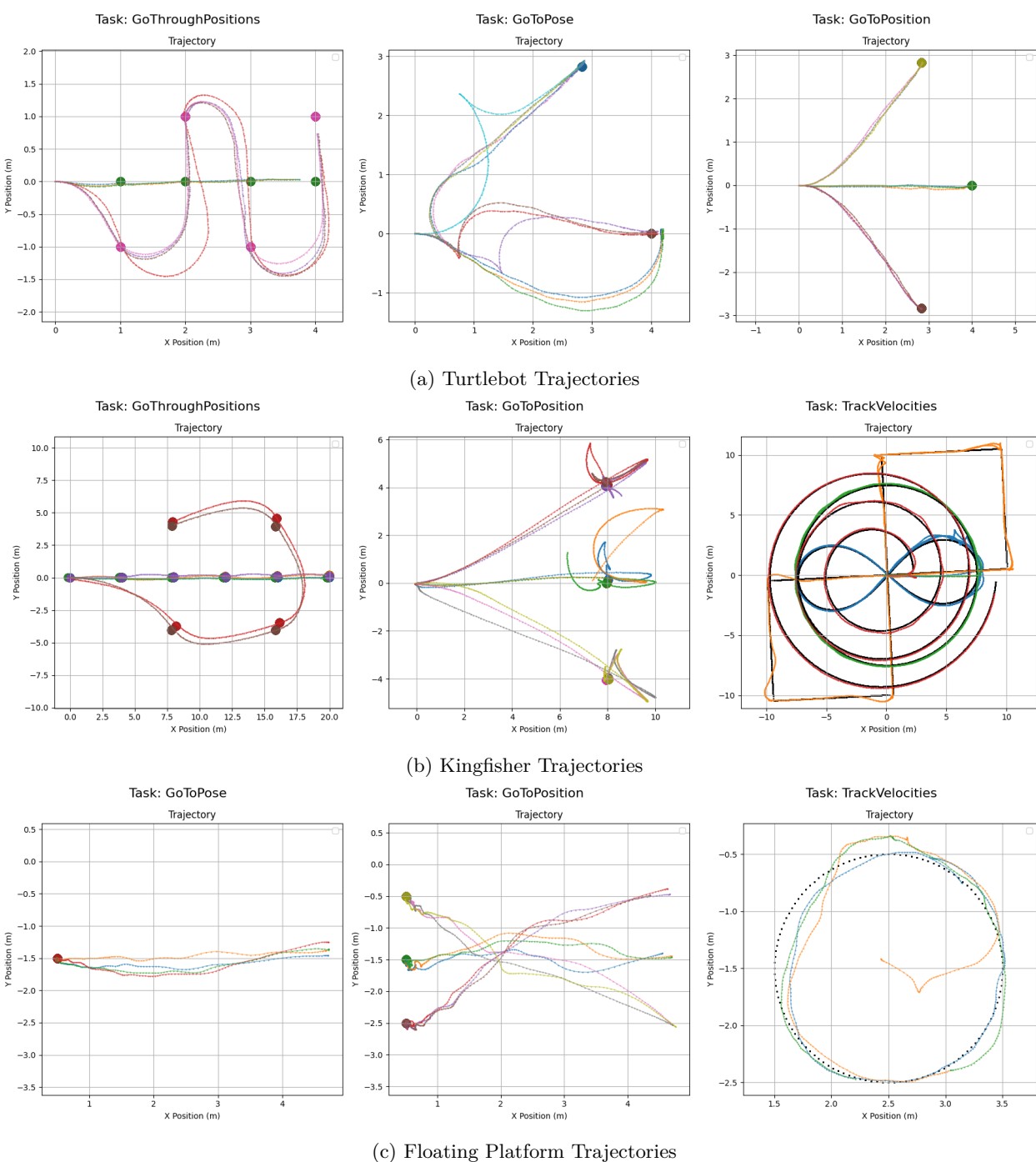

(a) Turtlebot Trajectories

(b) Kingfisher Trajectories

(c) Floating Platform Trajectories

Figure 9: Robot trajectories in real-world experiments. In all graphs, the colored dotted lines represent the robot trajectories, with different colors corresponding to different runs. For the tasks *GoThroughPositions*, *GoToPose*, and *GoToPosition*, the circles indicate the final targets. For *TrackVelocities*, the black dots denote target positions generated by a trajectory generator.

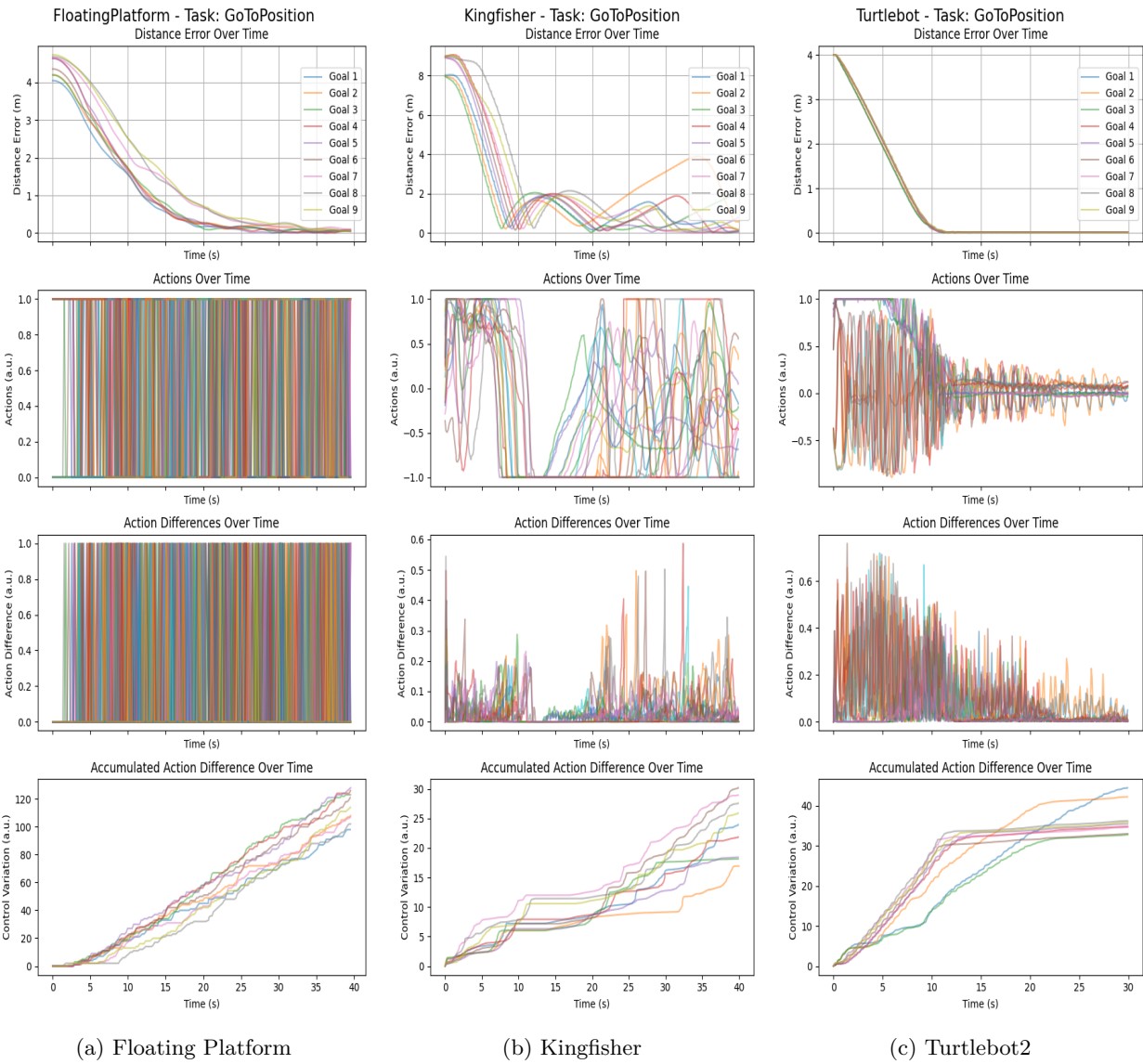

(a) Floating Platform          (b) Kingfisher          (c) Turtlebot2

Figure 10: Repeated field trials of the *GoToPosition* task across three robotic platforms. (10a) **Floating-Platform**: The metrics demonstrate consistent convergence, and the actions highlight the use of discrete thrust commands. The thrusters are engaged continuously to maintain position and prevent drift after reach the goal. (10b) **Kingfisher**: The metrics indicate that the policy often overshoots the goal, triggering corrective actions and suggesting potential improvements to the dynamics model to reduce the sim-to-real gap. (10c) **Turtlebot2**: The differential-drive system yields stable and repeatable trajectories with small final distance errors. In two experiments, it shows oscillating actions after reaching the target position, which indicates noisy position estimation.

## A.7 Customization of Environments

The framework is built on a modular and extensible architecture that significantly streamlines the process of adding new robots or tasks. This design eliminates the need to create a new environment from scratch for each addition. The core of this system is the use of factories within the environments directory, which dynamically generate the necessary components.

The file structure demonstrates this clear separation of concerns:

- robots: Contains the code for different robot models. To add a new robot, you simply create a new Python file (e.g., *robot_2.py*) that inherits from *robot_core.py*.

- tasks: Houses the definitions for various tasks. Adding a new task involves creating a new file (e.g., *task_2.py*) that extends *task_core.py*

```
1   source
2   | isaaclab_task
3   | | direct <-- Isaac Lab code
4   | | managed <-- Isaac Lab code
5   | | rans <-- Our Framework starts here
6   | | | domain_randomization
7   | | | environments
8   | | | | Single
9   | | | | | single_env.py <-- NO CHANGES NEEDED
10  | | | | | Agents
11  | | | | | | skrl
12  | | | | | | rl_games
13  | | | | | | rsl_rl
14  | | | robots
15  | | | | robot_core.py
16  | | | | FloatingPlatform.py
17  | | | | ...
18  | | | | new_robot_1.py <-- add here
19  | | | robots_cfg
20  | | | tasks
21  | | | | task_core.py
22  | | | | GoToPosition.py
23  | | | |...
24  | | | | new_task_1.py <-- add here
25  | | | tasks_cfg
26  | | | utils
```

Code 2: Folder structure of our framework integrated inside Isaac Lab.

This approach ensures that the fundamental environment code remains unchanged. By inheriting from a core class and adding the new component (be it a new task or a new robot) to its corresponding folder, the framework automatically integrates it. This not only reduces development time by needing to write less code but also minimizes the risk of introducing errors into the core environment.

Code 3 shows the structure of the environment with the API for the robot and tasks.

```
1
2   class SingleEnv(DirectRLEnv):
3
4       cfg: SingleEnvCfg
5
6       def _configure_gym_env_spaces(self):
7           """Configure the action and observation spaces for the Gym environment."""
8           # observation space (unbounded since we don't impose any limits)
9           super()._configure_gym_env_spaces()
10          self.single_action_space, self.action_space = self.robot_api.
    configure_gym_env_spaces()
11          self.actions = sample_space(self.single_action_space, self.sim.device, batch_size=
    self.num_envs, fill_value=0)
12
13      def _setup_scene(self):
14          self.robot = Articulation(self.robot_cfg.robot_cfg)
15          self.robot_api = ROBOT_FACTORY(
16              self.cfg.robot_name,
17              scene=self.scene,
18              robot_cfg=self.robot_cfg,
19              robot_uid=0,
20              num_envs=self.num_envs,
21              decimation=self.cfg.decimation,
22              device=self.device,
```

```
23            )
24            self.task_api = TASK_FACTORY(
25                self.cfg.task_name,
26                scene=self.scene,
27                task_cfg=self.task_cfg,
28                task_uid=0,
29                num_envs=self.num_envs,
30                device=self.device,
31            )
32
33            self.task_api.register_robot(self.robot_api)
34            self.task_api.register_sensors()
35
36            # add ground plane
37            spawn_ground_plane(prim_path="/World/ground", cfg=GroundPlaneCfg())
38            # clone, filter, and replicate
39            self.scene.clone_environments(copy_from_source=False)
40            self.scene.filter_collisions(global_prim_paths=[])
41            # add articultion to scene
42            self.scene.articulations[self.cfg.robot_name] = self.robot
43            # add lights
44            light_cfg = sim_utils.DomeLightCfg(intensity=2000.0, color=(0.75, 0.75, 0.75))
45            light_cfg.func("/World/Light", light_cfg)
46
47    def _pre_physics_step(self, actions: torch.Tensor) -> None:
48            self.robot_api.process_actions(actions)
49
50    def _apply_action(self) -> None:
51            self.robot_api.apply_actions()
52
53    def _get_observations(self) -> dict:
54            task_obs = self.task_api.get_observations()
55            observations = {"policy": task_obs}
56            return observations
57
58    def _get_rewards(self) -> torch.Tensor:
59            return self.task_api.compute_rewards()
60
61    def _get_dones(self) -> tuple[torch.Tensor, torch.Tensor]:
62            robot_early_termination, robot_clean_termination = self.robot_api.get_dones()
63            task_early_termination, task_clean_termination = self.task_api.get_dones()
64
65            time_out = self.episode_length_buf >= self.max_episode_length - 1
66            early_termination = robot_early_termination | task_early_termination
67            clean_termination = robot_clean_termination | task_clean_termination | time_out
68            return early_termination, clean_termination
```

Code 3: Demonstration of how the environment interacts with the robot and task components through the API. This dynamic instantiation ensures that the environment is not hard-coded for a specific robot or task. This modularity not only reduces development time but also enhances maintainability and robustness by isolating component-specific code from the core simulation loop.

### A.7.1 Algorithmic variation

In addition, thanks to the workflow inherited from Isaac Lab, users can easily select a different learning library or algorithm by modifying the configuration files rather than the code itself. As illustrated in Code 2, the `Agents` directory organizes all configuration files where training settings are specified. Swapping PPO for another algorithm (such as SAC or TRPO) simply involves changing a single entry in the corresponding YAML file, making the framework agnostic to the underlying learning implementation.

