# OpenReview forum: "RoboRAN: A Unified Robotics Framework for Reinforcement Learning-Based Autonomous Navigation"
_TMLR — Accepted by TMLR_

### Review · Reviewer_hTAR · 2025-09-10

**Summary Of Contributions:**

Summary: This paper proposes a multi-domain framework that enables the training, evaluation, and deployment of RL-based navigation policies across diverse robotic platforms and operational environments. It emphasizes the modular decoupling of robots and tasks. The authors verified the simulation training and real-world transfer capabilities of the policy on three platforms: the ground-based Turtlebot2, the surface-based Kingfisher, and the microgravity Floating Platform.

Strengths:
(1)The paper is well-structured and clearly organized, making it easy to follow the motivation, methodology, and experimental validation.
(2)The modular design is effective, as the framework decouples robots and tasks while supporting customizable dynamics and reward shaping. This design enables strong generalization and easy extensibility.
(3)The paper provides a lightweight, Dockerized deployment API with ROS2 integration, allowing policies trained in IsaacLab to be efficiently deployed on physical platforms. This demonstrates strong practical applicability.

Weaknesses:
(1)The selected tasks focus primarily on low-level control in obstacle-free environments, which limits the demonstration of RoboRAN’s extensibility to more complex scenarios such as visual perception or dynamic obstacle avoidance.
(2)Although the paper incorporates a domain randomization mechanism, it lacks a systematic ablation study to evaluate its effectiveness. For example, by comparing performance with and without different types of perturbations.

**Additional Comments:**

Please refer to the detailed comments above.

**Audience:**

Yes

**Audience Explanation:**

The paper addresses key challenges in reinforcement learning for robotics, namely generalization, sim-to-real transfer, and modular design. This is highly relevant to both academic researchers in embodied AI.

**Broader Impact Concerns:**

I do not identify any immediate ethical concerns arising from this work.

**Claims And Evidence:**

Yes

**Claims Explanation:**

The paper presents extensive simulation and real-world experiments across multiple robots and tasks, with detailed quantitative metrics. The modular design and sim-to-real transfer are well-demonstrated. However, the impact of domain randomization lacks ablation analysis, which slightly weakens the robustness claim.

**Requested Changes:**

(1) Ablation Study on Domain Randomization. The framework claims robustness via domain randomization, but no systematic analysis is provided to quantify its impact.
(2) Evaluation in More Complex Environments (e.g., with obstacles or perceptual inputs).

---

> ### Author Response · Authors · 2025-09-23
>
> We thank the reviewer for their positive and encouraging assessment. We particularly appreciate the recognition of our modular design, Dockerized deployment, and validation across multiple robot platforms.
>
> ### On Domain Randomization and Robustness Claims
> We agree with the reviewer that the impact of domain randomization (DR) was not evaluated through ablation studies in this submission. Our intention was not to claim novel techniques or rigorous analysis in this regard, but rather to adopt standard DR practices (e.g., noise injection in dynamics and sensors) to enhance sim-to-real transfer during training.
>
> In light of this feedback, we have rephrased the robustness claim throughout the paper (Sections 3.4 and Appendix A3) to clarify that:
> - DR is used as a practical tool to improve transferability.
> - We do not claim robustness as a core contribution or evaluate it systematically.
>
> We also briefly cite prior works that demonstrate the effectiveness of DR in similar contexts (e.g., in space robotics, legged locomotion, and aerial drones) to contextualize our use (Section 2, related works — paragraph 3 starting with *“Robustness”*).
>
> ### On Evaluation in Complex Environments
> We agree that evaluating policies in visually rich or obstacle-populated scenarios is an important next step. Our architecture and simulator backend already support obstacles in the scene, as well as Lidar or vision-based observations. Appendix A2 includes examples for the three robots trained on the **GoToPosition** task with obstacles, and Figures 6–8 show qualitative results for these examples in simulation.
>
> A comprehensive validation of all such tasks, including field tests, would be challenging to complete in a timely manner. Instead, we chose to focus on fundamental navigation tasks in relatively clean environments to ensure consistency across robots and provide reliable sim-to-real validation.

---

### Review · Reviewer_vMJe · 2025-09-17

**Summary Of Contributions:**

This paper presents RoboRAN, a unified and modular robotics framework designed to facilitate reinforcement learning (RL) across diverse robot platforms and tasks. Built on top of IsaacLab, RoboRAN enables interchangeable training of policies for multiple robot-task pairs within a shared and extensible infrastructure. The framework offers a consistent definition of robots and tasks, supporting reproducible experiments and scalable policy training. RoboRAN’s effectiveness is demonstrated through both simulation and real-world sim-to-real transfer experiments, involving a range of robotic platforms such as satellite simulators, unmanned surface vessels, and wheeled ground vehicles.

## Strengths:
- The multi-domain approach addressing terrestrial, aquatic, and space robots broadens applicability beyond typical RL frameworks.
- Real-world experiments validate the sim-to-real transfer capability, strengthening the practical relevance.
- Open-source release encourages reproducibility and community engagement.
- Uniform evaluation protocols improve benchmarking fairness and comparisons.

## Weaknesses
- It remains unclear how seamless and scalable the integration process is for even new robots and tasks
- The paper demonstrates training separate policies for each robot-task pair but does not justify the need for a unified framework if policies remain isolated. It lacks evidence that the framework enables knowledge sharing or transfer learning across different robots or tasks, which would better motivate its multi-domain approach and highlight novel contributions.
- Could the authors please elaborate on how the claimed interchangeability is demonstrated through training multiple navigation tasks across robots with distinct mobility systems (e.g., thrusters, wheels, water-based propulsion)? Specifically, it would be helpful to clarify whether this interchangeability involves shared policies, transfer learning, or simply training separate policies within the same framework.

**Audience:**

Yes

**Audience Explanation:**

TMLR's audience from robot learning and reinforcement leraning may be interested in this paper. However, the paper is quite application-driven, focusing heavily on robotics engineering and system integration, which might be more typical for robotics or control-focused venues. If the main contribution lies in software infrastructure and benchmarking rather than new RL algorithms or fundamental ML theory, some might see it as borderline for TMLR’s primary ML research scope.

**Claims And Evidence:**

Yes

**Claims Explanation:**

I think the authors have empirically support all their arguments and claims and I do believe that they have a huge contribution in robot learning community from application perspective. However, I am not convinced by the connection between the motivation and the proposed framework. From a practical standpoint, if a lab operates only one robot type, a dedicated platform for that robot may suffice, reducing the need for a unified framework. Conversely, when multiple robot types are involved, it’s unclear what benefit arises from having all robots solve the same task independently without collaboration or knowledge sharing. Without enabling inter-robot communication or transfer learning, the practical advantages of this multi-robot platform appear limited.

**Requested Changes:**

- Please clarify the specific contributions to reinforcement learning or machine learning beyond software infrastructure.
- Provide more details on the additional work and innovations compared to IsaacLab, especially focusing on RL/ML aspects and unique challenges addressed during training.
- Include more experiments that demonstrate the unique benefits and necessity of this unified framework. Specifically, identify problem settings or tasks where this framework is the only viable option, as training across multiple platforms individually may be inconvenient but not a critical barrier.

---

> ### Author Response · Authors · 2025-09-23
>
> We thank the reviewer for their constructive feedback and agree that the motivation and impact of our framework can be better clarified. Below we outline our contributions, target use cases, and how the framework complements algorithmic innovation.
>
> ### Clarifying Motivation, Scope, and Contributions
> The understanding that we train isolated policies for each robot–task pair is correct. Our framework allows researchers to benchmark and scale their algorithms across distinct robotic morphologies and control modalities by making task and reward design reusable across robots and robot interfaces compatible across tasks. By providing a validated, sim-to-real-compatible, and extensible multi-robot, multi-task environment, we offer a platform to test approaches in transfer or multitask learning algorithms.
>
> Beyond software infrastructure, we contribute:
> - A systematic demonstration of modular design that accommodates very different actuation systems, which is non-trivial in robotics.
> - Real-world experiments validating the task–robot separation and entire pipeline.
>
> We do not claim our platform is the only viable solution, nor that it replaces existing RL benchmarks. Rather, we position it as a practical and sim-to-real-validated option that fills a gap between overly simplified benchmarks and rigid, robot-specific environments.
>
> ### Why Adopt the Framework?
> - **Task–Robot Decoupling:** Users can re-use tasks, reward logic, and evaluation pipelines across different robots without redefining them per robot. This has been validated both in simulation and field deployments.
> - **Sim-to-Real Validity:** The same task structure used in simulation can be transferred to real-world deployment, preserving both reward and evaluation criteria while facilitating practical validation.
> - **Infrastructure for Future Research:** Our framework provides a clean foundation for integrating transfer learning, multitask training, or policy distillation approaches, which we explicitly identify as future research directions (see revised “Conclusions” section).
> - **Low Setup Barrier:** For labs with a single robot, the framework still provides an efficient starting point to test multiple tasks without writing robot-specific code for each. Conversely, for teams developing new RL methods, our setup enables robust evaluation across heterogeneous robots, even in simulation alone.
>
> ### On Seamless Integration and Scalability
> We kindly refer the reviewer to our reply to Comment 1 (Reviewer hMQx), where we describe the addition of an Appendix section (A7) with more details regarding this concern.
>
> ### Summary of Changes to the Paper Addressing This Review
> - Added (in the “Conclusions” section) a future work paragraph on multi-task and transfer learning possibilities.
> - Included implementation details and examples to show integration simplicity in Appendix A7.

---

### Review · Reviewer_hMQx · 2025-09-22

**Summary Of Contributions:**

This paper proposes a new open-source framework for benchmarking robotic navigation acrosscombinations of environment modalities (land, water, space) and tasks (position, pose, waypoints, velocities). A single reward is proposed to unify different task types, such that any policy for a chosen robot-task can be trained using that reward. The framework supports domain randomization to enable training and deploying policies for sim-to-real transfer.

**Audience:**

Yes

**Audience Explanation:**

There is wide interest in RL and robotics.

**Claims And Evidence:**

No

**Claims Explanation:**

1. The paper claims "Our framework separates robot and task definitions using standardized APIs, enabling fast experimentation with new robot–task pairings and facilitating the extension to novel platforms and navigation tasks". The fact that this paper demonstrated three different motion modalities (thrusters, wheels, water propulsion) is good but not sufficient to support the claim that extending to a new platform is "fast". As currently presented, the paper does not show how difficult it is to use the framework to add a new task or customize/modify/add some new motion modality or dynamics. This is just a writing issue with the paper: I am not doubting the possibility of such extension, but rather just pointing out that the claim is not supported. One would expect something along the lines of "it took one hour of human effort to add a new task", or "here is the API, see how easy it is to customize" (this can be in the appendix. The linked code is not final so reviewers are not expected to read it).
2. Only PPO was run. Yes, comparing algorithms is not the point, but at least show that your framework allows training and deploying using various algorithms. Running some would be good (who knows, maybe there is high sensitivity to algorithms and need tuning, then your framework would have to support tuning). Otherwise, showing some snippets of the API in the appendix may be enough.

**Requested Changes:**

1. Address the above comment about the unsupported claim of ease of customization/adding new robots/tasks

---

> ### Author Response · Authors · 2025-09-23
>
> Thank you for your review and constructive suggestions. Our response to the two main concerns is as follows:
>
> ### 1. Claim about “fast experimentation,” APIs, and extensibility
> We agree that the original wording was vague. In the revision we:
>
> - Add **Appendix A7**, which makes the interfaces explicit. It includes:
>   - A concise listing of the **Task interface** (state specification, termination, reward composition, reference trajectory where relevant) and **Robot interface** (actuation, dynamics step, observation mapping, reset).
>   - A short, self-contained example showing how to add a new task to an existing robot, or add a new robot to an existing task. For each step we indicate the files to create or extend and the minimal methods to implement.
>
> - Replace the original claim with a more precise statement about modularity:
>   > *“Our framework separates robot and task definitions using standardized APIs, {\color{blue}{minimizing integration overhead and enabling new robot–task pairings without altering existing modules}}.”*
>
> ---
>
> ### 2. Only PPO was run and algorithm support
> We understand the concern about demonstrating only one algorithm. To address this point, in the revision we:
>
> - Add **Appendix A7** (with Code2) explaining how to swap the learning backend or algorithm, and illustrating in the folder structure where the configurations for the desired library and algorithm can be set up.
>
> - Clarify that analyzing sensitivity to different algorithms is of interest, but in practice requires tuning algorithm-specific hyperparameters. Because this modularity is already provided by the standard Isaac Lab implementation, we prioritized reporting results for the **robot–task abstraction**.

---

### Decision · Action_Editor_6Z5k · 2025-11-03

**Recommendation:** Accept with minor revision

**Additional Comments:**

The reviewers agree that RoboRAN represents a technically sound and practically valuable system contribution. It offers a unified infrastructure that lowers the entry barrier for RL-based navigation research and provides a scalable foundation for future work in multi-robot and transfer learning. The paper is well-written, clearly structured, and reproducible, with a comprehensive suite of experiments and an open-source release.
While the contribution is primarily framework-oriented rather than algorithmic, it is judged to be an important enabler for the robotics and RL research communities. The authors have satisfactorily addressed reviewer concerns in the revision, particularly by clarifying framework modularity, adding implementation examples (Appendix A7), and refining claims about robustness and modularity.

However, for the final version of the paper I request the following minor changes to the manuscript in its current form:
- please make the relationship to IsaacLab clearer (maybe with a focus on unique RL-oriented features); a more explicit-comparison to IsaacLab’s baseline capabilities makes the contribution of RoboRAN clearer

**Audience:**

Yes

**Audience Explanation:**

This submission is of particular interest to researchers in RL for robotics, embodied AI, and sim-to-real. While the work is more system- and framework-oriented, it provides a valuable research infrastructure that directly supports experimental reproducibility and scalable evaluation of RL-based robotics. RoboRAN is a useful tool for both applied robotics researchers and machine learning practitioners interested in generalization and transfer across diverse physical platforms. Overall, RoboRAN may serve as an enabler for experimental rigor in RL-based robotics.

**Claims And Evidence:**

Yes

**Claims Explanation:**

After some discussion with the reviewers the claims are now well-supported by either empirical or implementation (i.e. adding new robots or tasks) evidence: The authors demonstrate sim-to-real transfer across three distinct robotic systems, provide detailed modular design documentation, and include appendices illustrating integration simplicity and task-robot interchangeability. The presented experiments and code release offer sufficient evidence to substantiate the central claims regarding modularity, scalability, and practical applicability of RoboRAN.